# Changes in Skin Cancer-Related Behaviors, Distress, and Beliefs in Response to Receipt of Low- to Moderate-Penetrance Genetic Test Results for Skin Cancer Risk

**DOI:** 10.3390/cancers16234027

**Published:** 2024-11-30

**Authors:** Monica Khadka, John Charles A. Lacson, Steven K. Sutton, Youngchul Kim, Susan T. Vadaparampil, Brenda Soto-Torres, Jennifer L. Hay, Peter A. Kanetsky

**Affiliations:** 1Morsani College of Medicine, University of South Florida, Tampa, FL 12901, USA; mkhadka1@usf.edu; 2Department of Cancer Epidemiology, H. Lee Moffitt Cancer Center and Research Institute, Tampa, FL 33162, USA; 3Department of Biostatistics and Bioinformatics, H. Lee Moffitt Cancer Center and Research Institute, Tampa, FL 33612, USA; steve.sutton@moffitt.org (S.K.S.); youngchul.kim@moffitt.org (Y.K.); 4Department of Health and Behavioral Outcomes, H. Lee Moffitt Cancer Center and Research Institute, Tampa, FL 33612, USA; susan.vadaparampil@moffitt.org; 5Public Health Program, Ponce Health Sciences University, Ponce, PR 00716, USA; bsoto@psm.edu; 6Department of Psychiatry and Behavioral Sciences, Memorial Sloan Kettering Cancer Center, New York, NY 10017, USA; hayj@mskcc.org

**Keywords:** randomized trial, intervention trial, melanoma, prevention, *MC1R*, precision prevention, genetic testing, public health genetics

## Abstract

There is a proliferation of precision approaches to cancer prevention for which personalized cancer risk information is used to motivate protective behaviors. Exploration of this potential is understudied in the skin cancer context. We examined changes in skin cancer-related behaviors, distress, and beliefs over time after *MC1R* genetic testing and receipt of average-risk or higher-risk personalized feedback in 568 non-Hispanic White and 463 Hispanic participants. We found those receiving higher-risk feedback initially had reduced weekend hours in the sun and modestly increased skin cancer worry, both of which dissipated over time. Those receiving higher-risk feedback reported a durable increased perceived risk of skin cancer compared with those receiving average-risk feedback. These results confirm the positive motivational impact of precision prevention on short-term behavior and durable impacts on skin cancer beliefs that sustain such behavior change.

## 1. Introduction

Public health genomics seeks to translate findings from the genomic sciences to inform population health in an effective and responsible fashion, which includes the examination of benefits, harms, and safety issues associated with the receipt of genomic information [1,2,3]. The benefits of a genetics-based precision prevention approach include enhancing motivation for the adoption of behavior modifications toward healthier choices, ensuring greater personal control over health choices and outcomes, and the engagement of the wider family system in precision prevention through family member genetic testing, i.e., cascade testing [4]. A variety of potential harms can result from a genetics-based precision prevention approach, and these harms must be mitigated. In particular, receiving information that one is at higher genetic risk can be distressing, while receiving information that one is at average genetic risk can result in adopting a false sense of security [5]. Although excessive distress can be a potential harm resulting from precision prevention, some amount of concern about disease risk can be motivating and result in positive behavior changes [6,7,8]. As such, research to examine benefits and harms is warranted to ensure best practices for the use of genomic information by the general public.

For skin cancer, previous studies have assessed the benefits and harms of genetic testing for cyclin-dependent kinase inhibitor 2A (*CDKN2A*)—the major high-penetrance gene for melanoma [9,10]—among members of melanoma-prone families. These studies have noted public health benefits of testing and provision of risk feedback, spanning healthy changes in psychosocial, behavioral, and cognitive measures for all tested individuals, as well as specific and differential improvements in prevention and screening behaviors for individuals who test positive for a pathogenic mutation. These improvements also extend to family members of those who test mutation positive [11,12]. Outside of the high-risk setting, a small number of studies, including three of our own, have assessed outcomes associated with the receipt of low- to moderate-penetrance genetic testing for skin cancer risk, including melanoma [13,14,15,16]. These studies engaged individuals drawn from the general population to address primary and secondary prevention outside of the context of familial aggregation. Findings from these studies showed promising improvement in some prevention behaviors with no indication of increased cancer distress among those receiving higher-risk genetic information or the adoption of increased risk behaviors, such as outdoor tanning, that could reflect a false sense of security among those receiving genetic information conveying average or lower risk, when compared with non-genetics-based intervention control groups.

There are numerous genetic variants at the single coding exon of the melanocortin-1 receptor (*MC1R*) gene [17,18]. Many *MC1R* variants confer an elevated risk of melanoma, squamous cell carcinoma, and/or basal cell carcinoma, and a large proportion of individuals of European ancestry carry at least one higher-risk *MC1R* variant [19,20]. In our two prior efficacy studies [13,14], we compared the receipt of skin cancer precision prevention materials incorporating *MC1R* risk with the receipt of publicly available non-genetics-based skin cancer prevention materials to determine changes in skin cancer prevention activities at two post-intervention time points and to assess potential psychosocial mediators and moderators of the intervention effect. Participants were block randomized within average- and higher-risk *MC1R* categories. However, our published main analyses did not directly compare changes among participants at average and higher *MC1R* risk. Notably, this comparison will inform the expected shorter- and longer-term outcomes associated with feedback receipt in real-world settings where *MC1R* genetic testing may become available.

In the secondary data analyses reported here, we compared the effects of the precision prevention on skin cancer-related behaviors, distress, and beliefs, among participants at *MC1R* higher risk and *MC1R* average risk at the first of two follow-up surveys. We then evaluated the durability of any statistically significant difference by comparing the effects at the second follow-up.

## 2. Materials and Methods

### 2.1. Setting and Participants

Information on the setting, participants, research surveys and study measures, genetic testing, and precision prevention materials of the two parent intervention studies, one among non-Hispanic White (NHW) participants and one among Hispanic participants, has been described in detail elsewhere [13,14]. Participants who self-reported being non-Hispanic, White, fluent in English, and 18 years of age or older and who reported having few skin cancer risk phenotypes based on ability to tan, tendency to burn, hair color, and freckling were recruited from primary care clinics in Tampa, Florida, between September 2015 and September 2018. NHW individuals who had a full-body skin examination within the past year or a personal history of melanoma were ineligible for study participation. Participants who self-reported being Hispanic, fluent in either English or Spanish, and 18 (Tampa) or 21 (Ponce) years of age or older were recruited from primary care settings in Tampa, Florida, and Ponce, Juana Díaz, and Salinas, Puerto Rico, between September 2018 and January 2020. Hispanic individuals who had a full-body skin examination within the past year, a personal history of melanoma, or a personal history of more than one basal cell carcinoma and/or squamous cell carcinoma were ineligible for study participation.

Although the two intervention studies were otherwise methodologically similar, the prevention emphasis differed. The trial among NHW participants exclusively focused on the prevention of melanoma, while the trial among Hispanic participants focused on the prevention of skin cancer in general, including basal cell carcinoma and squamous cell carcinoma in addition to melanoma. This shift reflected the lower incidence of melanoma among Hispanic compared with NHW individuals and the elevated risk of basal cell carcinoma and squamous cell carcinoma conferred by *MC1R* genetic variants [20,21]. For simplicity, hereafter we refer only to “skin cancer” recognizing this term corresponds only to melanoma for NHW participants but to basal cell carcinoma, squamous cell carcinoma, and melanoma for Hispanic participants.

The intervention trial among NHW participants was approved by the Institutional Review Board of the University of South Florida (Pro00020044, approved on 2 February 2015), and the intervention trial among Hispanic participants was approved by the Institutional Review Boards of the University of South Florida (Tampa, FL, USA; Pro00020044, approved 30 August 2018), Ponce Health Sciences University (Ponce, PR, USA; 170807-BS, approved 6 December 2017), and the Comité de Seguimiento de la Investigación Clínica at Hospital Damas (HD 19-17, approved 18 December 2017). Written informed consent was obtained from all participants.

### 2.2. Randomization

Participants were genotyped for naturally occurring variation at *MC1R* and were classified as average risk, i.e., variants conferring less than 80% increased odds of skin cancer, or higher risk, i.e., variants conferring at least 80% increased odds of skin cancer, as previously described [13,14]. Participants randomized to the precision prevention arm received mailed materials that provided *MC1R* risk feedback and ways to prevent skin cancer set in the context of inherited genetics. Participants randomized to the standard arm were mailed publicly available information about skin cancer prevention; these individuals did not contribute to the analyses described here. A member of the research team called participants within two weeks of the mailing to assure the receipt of the study materials.

### 2.3. Data Collection

Participants completed a baseline survey prior to receiving skin cancer precision prevention materials. This survey assessed age, sex, race, Hispanic identity, marital status, education, outdoor work, health literacy [22], health numeracy [23], untanned skin color, language preference, family history of melanoma, family history of squamous cell or basal cell skin cancer, and family history of other cancers. The baseline survey also solicited information on skin cancer-related behaviors over the past year and on skin cancer-related distress and beliefs.

After receipt of skin cancer precision prevention materials, participants completed two additional surveys: at 6 and 12 months for NHW participants and at 3 and 9 months for Hispanic participants. The two surveys solicited information on skin cancer-related behaviors over the prior study period and on skin cancer-related distress and beliefs, allowing for the evaluation of change since the baseline.

### 2.4. Skin Cancer-Related Behaviors

Information on skin cancer-related behaviors was collected using a standardized survey. We assessed the time spent outside (in hours) from 10 a.m. to 4 p.m. on weekdays and weekends separately, the frequency of outdoor intentional tanning (never, rarely, sometimes, often, or always), the number of sunburns, and the number of tanning bed sessions. We solicited information about the frequency (never, rarely, sometimes, often, or always) of sun protection behaviors including sunscreen usage, wearing of sunglasses, wearing of a hat, wearing of a shirt that covers the shoulders, and seeking shade or using and umbrella while outside. In addition to examining the frequency of these behaviors individually as outcomes, we also created a summary variable to indicate the total number of sun protection behaviors practiced “often” or “always” (range 0–5).

### 2.5. Skin Cancer-Related Distress

Measures of distress included worry, concern, and more severe distress, measured as intrusive and avoidant ideation about skin cancer. General skin cancer worry was measured using a 3-item skin cancer adaptation of Lerman’s cancer worry scale (range 1–5) [11,24]. Recent worry and concern about skin cancer over the past two weeks were assessed by asking “During the past two weeks, how often have you worried about the possibility of getting skin cancer?” (rarely or never, sometimes, often, or all of the time) and “During the past two weeks, how concerned have you been about the possibility of getting skin cancer?” (not at all concerned, a bit concerned, concerned, or very concerned). Recent intrusive and avoidant ideation about skin cancer over the past week was measured using the 15-item Impact of Events scale (range 0–75) [25].

### 2.6. Skin Cancer-Related Beliefs

Measures of beliefs about skin cancer included the assessment of perceived risk, perceived severity, response efficacy, and self-efficacy. Perceived risk included two established measures. The perceived absolute risk of developing skin cancer was measured as “Do you think you are likely or unlikely to get skin cancer?” (likely, unlikely, or no idea), and the perceived comparative risk of developing skin cancer was measured as “Compared to the average person of your age and gender, what is the chance that you will develop skin cancer in the future?” (well below average, below average, average, above average, or well above average) [26,27]. Perceived skin cancer severity was measured using the average of a 7-item 4-point Likert-type scale [26]. Response efficacy was measured using seven preventive activities (e.g., wearing long-sleeved shirts and using sunscreen of SPF 15 or higher) and asking participants how important each activity was to reduce skin cancer risk (range 1–4), while self-efficacy was measured by asking how capable they were to perform these activities (range 1–4).

### 2.7. Statistical Analysis

Because the timing of follow-up surveys differed by study population, all analyses were conducted separately among NHW participants and Hispanic participants. First, we evaluated differences in demographics, skin cancer-related prevention behaviors, distress, and beliefs reported at the baseline between individuals at *MC1R* average risk and *MC1R* higher risk using Student’s *t*, Wilcoxon, or chi-squared tests, as appropriate. All baseline variables with a difference surpassing the significance threshold of *p* ≤ 0.10 were included in models evaluating change. We first evaluated the change from the baseline to the first follow-up. Then, statistically significant differences were evaluated for the change from the baseline to the second follow-up to gauge the durability of the effect.

For each outcome measured at the first follow-up (i.e., at six months among NHW participants and at three months among Hispanic participants), multivariable regression was used to estimate the effects of receiving *MC1R* higher-risk feedback with the following covariates: (1) the outcome measured at the baseline, (2) the predictors of missingness at the first follow-up, (3) the predictors of the outcome at the first follow-up, and (4) demographic variables that differed by risk group at the baseline. Linear regression evaluated the following outcome measures: weekday and weekend sun exposure, the number of sunburns, frequency of outdoor intentional tanning, the total number of sun protection behaviors, the skin cancer worry scale, intrusive and avoidant skin cancer ideation, the comparative risk of developing skin cancer, response efficacy, self-efficacy, and perceived severity. Logistic regression evaluated the following outcome measures: wearing a hat, wearing a shirt with sleeves, wearing sunglasses, wearing sunscreen, and seeking shade or using an umbrella, each dichotomized at always/often versus sometimes/rarely/never; recent worry about skin cancer, dichotomized at rarely/never versus sometimes/often/all the time; and recent concern about skin cancer, dichotomized at not at all versus a bit/ concerned/very. A multinomial regression model was used to evaluate the absolute risk of developing skin cancer.

Outcomes that demonstrated a statistically significant group difference at the first follow-up were evaluated for group differences at the second follow-up, i.e., at 12 months for NHW participants and at nine months for Hispanic participants. These analyses paralleled those described above.

R software (ver. 4.1.0, R Foundation for Statistical Computing, Vienna, Austria, RRID:SCR_001905), RStudio (ver. 1.4.1717, RStudio Team, Boston, MA, USA, RRID:SCR_000432), and SAS (ver. 9.4., Statistical Analysis System, Fargo, ND, USA, RRID:SCR_008567) were used to conduct analyses.

## 3. Results

A total of 226 NHW participants at *MC1R* average risk and 342 at *MC1R* higher risk completed the baseline survey. For Hispanic participants, the baseline completion numbers were 195 and 268 for *MC1R* average risk and higher risk, respectively. There were few statistically significant baseline differences in demographics, skin cancer-related behaviors, skin cancer-related distress, or skin cancer-related beliefs across participants who received *MC1R* higher- versus average-risk feedback (Table 1). Among NHW participants, those at average risk had greater educational attainment (*p* = 0.02) and reported higher levels (*p* = 0.02) of self-efficacy compared with higher-risk participants. NHW participants at *MC1R* average risk also tended to have a somewhat darker untanned skin color (*p* = 0.10), engaged in indoor tanning over the past year (*p* = 0.05), and were less likely to use sunscreen often or always over the past year (*p* = 0.07). Among Hispanic participants, those at average risk reported slightly higher levels of self-efficacy (*p* = 0.01) than did participants at higher risk. Hispanic participants at *MC1R* average risk also tended to report slightly greater response efficacy (*p* = 0.08) and were more likely to complete the telephone follow-up call after the receipt of precision prevention materials (*p* = 0.06).

### 3.1. NHW Participants, MC1R Higher Risk Versus Average Risk

Adjusted associations of intervention effects at the first follow-up on skin cancer-related behaviors, skin cancer-related distress, and skin cancer-related beliefs comparing NHW participants at *MC1R* higher versus average risk are shown in Table 2. The higher-risk group exhibited a greater decrease in average weekly weekend hours compared with those at average risk (β = −0.25; 95% CI, −0.46–[−0.04]). The higher-risk group also tended toward improvements for seven of the remaining nine skin cancer prevention behaviors compared with the participants at average risk. Although average levels of skin cancer-related distress were low, the higher-risk group had a higher average on the skin cancer worry scale (β = 0.09; 95% CI 0.01—0.18) than did the average-risk group. There were several statistically significant differences in skin cancer-related beliefs. The higher-risk group had a higher average perceived risk of developing skin cancer than did the average-risk group, including both when reporting a relative risk compared with others like them (β = 0.40; 95% CI 0.24—0.57) and in absolute terms (no idea vs. unlikely, OR = 1.73; 95% CI 1.04—2.87; likely vs. unlikely, OR = 2.16; 95% CI 1.10—4.22). The higher-risk group also had a higher average response efficacy score (β = 0.10; 95% CI 0.01—0.19) compared with participants in the average-risk group.

At the second follow-up survey, i.e., at 12 months, only intervention effects for skin cancer-related beliefs persisted among NHW participants at *MC1R* higher risk (Table 3). Those at higher risk continued to report a greater absolute risk (likely vs. unlikely, OR = 2.25; 95% CI 1.35—3.75) of developing skin cancer compared with the average-risk group, and on average, those at higher risk also continued to report a greater comparative risk (β = 0.49; 95% CI 0.33—0.65) of developing skin cancer compared with the average-risk group. Moreover, the higher-risk group continued to be more likely to report having “no idea” about their absolute risk of developing skin cancer (vs. unlikely, OR = 3.35; 95% CI 1.70—6.62) compared with the average-risk group.

### 3.2. Hispanic Participants, MC1R Higher Risk Versus Average Risk

Adjusted associations of intervention effects at first assessment, i.e., at three months, on skin cancer-related behaviors, distress, and beliefs comparing Hispanic participants at *MC1R* higher versus average risk also are shown in Table 2. There were no differences in changes for skin cancer-related behaviors or skin cancer-related distress, and there was no clear overall tendency toward improved skin cancer-related behaviors in those at higher risk. However, the higher-risk group had an elevated perceived risk of developing skin cancer. On average, the higher-risk group reported a greater comparative risk (β = 0.50; 95% CI 0.28–0.73) of developing skin cancer compared with the average-risk group. The higher-risk group also reported a greater absolute risk (likely vs. unlikely, OR = 3.40; 95% CI, 1.40–8.27) of developing skin cancer compared with the average-risk group.

At the second follow-up, intervention effects on skin cancer-related beliefs persisted among Hispanic participants at *MC1R* higher risk (Table 3). On average, the higher-risk group continued to report a greater comparative risk of developing skin cancer (β = 0.42; 95% CI 0.17–0.67) compared with those at average risk. Although differences in the absolute risk of developing skin cancer also persisted, the higher-risk group now more often reported having “no idea” (vs. unlikely, OR = 2.22; 95% CI, 1.07–4.60) of their absolute risk of developing skin cancer at nine months; the intervention effect on reporting likely (vs. unlikely) to get skin cancer remained elevated but was no longer statistically significant.

## 4. Discussion

The assessment of low- to moderate-penetrance precision prevention across a broad range of constructs is understudied yet needed to accurately estimate the trade-off between its benefits and harms. The findings from our analyses highlight several key points about the psychosocial and behavioral impact in diverse individuals receiving results of *MC1R* genetic testing. We found several differences between higher- and average-risk participants in the expected directions for selected measures of skin cancer-related behaviors, skin cancer-related distress, and skin cancer-related beliefs, although only the difference in beliefs, namely, skin cancer risk perceptions, persisted over the course of the study. Of note, few differences were noted across NHW and Hispanic participants.

Our most robust study findings related to the impact of *MC1R* testing on risk perceptions with participants at *MC1R* higher risk reporting higher absolute risk and greater comparative risk for developing skin cancer compared with those at average risk. Not only did the elevated perceived risk among higher-risk participants persist over time, but an elevated perceived risk among higher-risk participants also was evident in both the Hispanic and NHW study populations. These findings confirm the impact of *MC1R* precision prevention on this key mediator of behavioral adoption [28,29]. Nonetheless, the modest impact of *MC1R* precision prevention on changes in skin cancer-related behavior in our analyses highlight the need for additional content, such as the development of an action plan [30], to aid the translation of heightened risk perception to actual behavior changes.

Another noteworthy observation regarding risk perception is participants’ response of “no idea” to the question about the perceived absolute risk of skin cancer despite receiving materials that clearly conveyed their genetic risk. Importantly, a lack of health literacy can account for responses indicating high levels of uncertainty (e.g., no idea, do not know) among research participants [31]. To explore the possible impact of health literacy on the perceived risk of developing skin cancer at the first follow-up, we reran models for NHW participants adjusting for health literacy, and the regression estimates did not substantively change. Health literacy was already included in models for Hispanic participants (as a predictor of follow-up survey missingness [13]); however, when health numeracy was further added to regression models, the point estimate for “no idea” (of my absolute risk of skin cancer vs. “unlikely”) at the first follow-up became stronger and statistically significant, and the point estimate for “likely” also became stronger. Responses of “no idea” may also indicate a level of health avoidance or reluctance to engage with health-related information that could potentially cause distress [31]. For example, in our NHW efficacy study, participants at average risk who did not complete the post-intervention telephone call with the study team were less likely to accurately recall their *MC1R* risk category, and the lack of telephone call completion could be a behavioral indicator of health information avoidance. Our findings point to the value of including a “no idea” response when assessing risk perception in diverse populations. They also suggest the need to deploy strategies to engage and support people exhibiting health avoidance to maximize the impact of genetic risk testing across all populations.

Considering behavioral impacts, our findings indicated a decrease in weekend hours spent outdoors among those at *MC1R* higher risk compared with those at *MC1R* average risk. We note that weekend hours are likely more fully in participants’ control than hours spent outdoors during the week. This intentional effort to reduce sun exposure conveys that participants at higher genetic risk are responsive to genetic testing information provided to them. We also found general trends toward improved sun protection, e.g., greater report of seeking shade or using an umbrella often or always while outdoors, among participants at higher risk. Interestingly, although NHW participants at *MC1R* higher risk already reported statistically less indoor tanning and more frequent use of sunscreen at the baseline than did NHW participants at *MC1R* average risk, we still observed a positive intervention effect on these two skin cancer prevention behaviors comparing these two risk groups.

In other work, skin cancer genomic feedback comparing responses with the receipt of higher versus low/average risk derived from a polygenic risk score for melanoma resulted in increased sun protection behavior, particularly sunscreen use [15]. Intervention effects from the current study were not statistically significant and likely reflect improvements in most skin cancer-related behaviors, including reduced sun exposure and increased sun protection, for both average- and higher-risk groups and in both Hispanic and NHW participants, as we have previously shown [13,14].

Reported levels of skin cancer-related distress were consistently low across risk feedback type, follow-up time, and study setting, and our analysis revealed little evidence of troubling distress associated with the receipt of higher-risk genetic information. Although we observed a small increase in several distress measures among NHW participants receiving higher-risk information, only cancer worry attained statistical significance at the initial follow-up, but this finding did not persist over time. Consistent with prior work [15,16], overall, our findings on skin cancer-related distress provide further evidence that feedback of low- to moderate-penetrance genetic information results in minimal psychological harms.

Although the two parent intervention studies, which provide the data on participants receiving *MC1R* genetic feedback analyzed here, were robustly designed to allow powerful examination of small effect sizes of the receipt of skin cancer precision prevention materials compared with the receipt of standard, non-genetic, skin cancer materials, the secondary analyses we presented here did not follow on the original randomization scheme. Fortunately, there were limited differences in baseline measures between analytic groups, and all differences were accounted for in multivariable regression analyses. The distinction in the timing of follow-up surveys between the two parent studies, which resulted from differences in their respective duration of funding, precluded a formal pooled analysis. It is possible that receiving risk feedback focused only on melanoma (NHW participants) can evoke different psychosocial and behavioral responses than receiving risk feedback focused on melanoma, squamous cell carcinoma, and basal cell carcinoma (Hispanic participants). Moreover, the response to risk feedback also is likely intertwined with aspects of differential skin cancer incidence and lethality for the two participant samples. As previously noted elsewhere [13,14], the completion rates of the follow-up surveys were modest (48–77%). Although predictors of survey missingness were incorporated into multivariable models, the decrease in the number of participants with longitudinal analyzable data could have impacted the power to detect effect differences across *MC1R* risk groups.

## 5. Conclusions

We evaluated how skin cancer-related behaviors, skin cancer-related distress, and skin cancer-related beliefs were impacted by the knowledge of *MC1R* genetic risk status, which provides insight into the potential benefits and harms that can accompany low to moderate genetic testing. Participants receiving *MC1R* higher-risk feedback reported a durable increased skin cancer perceived risk, although decreases in sun exposure hours were not sustained over the study period. As protection motivation theory, which grounded the development of our precision prevention materials, theorizes that perceived risk is an important mediator of behavior change, additional intervention reinforcement of some type may be needed to promote longer-term improvements in prevention behaviors. Skin cancer distress was low, and participants at *MC1R* higher risk only reported slight heighted skin cancer worry that dissipated over the course of the study. The findings from our analyses suggested the benefits of low to moderate genetic testing for skin cancer prevention outweigh the harms. This study provides important information vital to advancing public health genomics.

## Figures and Tables

**Table 1 cancers-16-04027-t001:** Baseline characteristics of the two study populations by *MC1R* risk category.

	Non-Hispanic White			Hispanic		
	Average Risk	Higher Risk		Average Risk	Higher Risk	
Characteristic	(*n* = 226)	(*n* = 342)	*p*	(*n* = 195)	(*n* = 268)	*p*
**Demographics**						
Age in years (mean, SD)	46.6 (16.8)	47.7 (14.9)	0.40	45.9 (14.9)	44.0 (15.5)	0.11
Female (*n*, %)	116 (51.3)	168 (49.1)	0.67	134 (68.7)	186 (69.4)	1.00
Hispanic identity (*n*, %)			n/a ^b^			0.68 ^c^
Puerto Rican	0	0		131 (67.2)	174 (64.9)	
Central or South American (not Brazilian)	0	0		14 (7.2)	40 (14.9)	
Cuban	0	0		22 (11.3)	17 (6.3)	
Mexican	0	0		4 (2.1)	19 (7.1)	
Dominican Republic	0	0		8 (4.1)	7 (2.6)	
Mixed	0	0		9 (4.6)	7 (2.6)	
Other	0	0		7 (3.6)	4 (1.5)	
Race (*n*, %)			n/a ^b^			0.78
White	226 (100)	342 (100)		152 (77.9)	213 (79.5)	
Other	0	0		43 (22.1)	55 (20.5)	
Marital status (*n*, %)			0.48			0.64
Single or never married	60 (26.5)	79 (23.1)		51 (26.2)	75 (28.0)	
Married, domestic partnership, or civil union	134 (59.3)	201 (58.8)		112 (57.4)	143 (53.4)	
Divorced, separated, or widowed	32 (14.2)	59 (17.3)		30 (15.4)	48 (17.9)	
Education (*n*, %)			**0.02**			0.13
Less than high school or GED	66 (29.2)	108 (31.6)		18 (9.2)	40 (14.9)	
High school or GED	61 (27.0)	122 (35.7)		29 (14.9)	45 (16.8)	
Some college ^a^	49 (21.6)	53 (15.5)		57 (29.2)	73 (27.2)	
Four-year college degree	41 (18.1)	45 (13.2)		63 (32.3)	61 (22.8)	
Graduate degree or higher	9 (4.0)	10 (2.9)		26 (13.3)	42 (15.7)	
Family history of melanoma (*n*, %)	47 (20.8)	74 (21.6)	0.90	26 (13.3)	36 (13.4)	1.00
Family history of squamous cell or basal cell carcinoma (*n*, %)	61 (27.0)	99 (28.9)	0.71	10 (5.1)	12 (4.5)	0.92
Family history of other cancers (*n*, %)	145 (64.2)	215 (62.9)	0.80	112 (57.4)	140 (52.2)	0.34
Worked outdoors (*n*, %)	139 (61.5)	209 (61.1)	1.00	70 (35.9)	104 (38.8)	0.60
Health literacy (*n*, %)			0.83			0.78
Not at all, a little bit, or somewhat confident	25 (11.1)	25 (7.4)		29 (14.9)	49 (18.3)	
Quite a bit confident	47 (20.9)	85 (25.0)		65 (33.3)	78 (29.1)	
Extremely confident	153 (68.0)	230 (67.6)		99 (50.8)	136 (50.7)	
Health numeracy (*n*, %)			0.41			0.68
Very easy	92 (40.7)	142 (41.5)		52 (26.7)	76 (28.4)	
Easy	110 (48.7)	178 (52.0)		106 (54.4)	144 (53.7)	
Hard or very hard	23 (10.1)	19 (5.6)		35 (17.9)	45 (16.8)	
Untanned skin color (*n*, %)			**0.10**			0.61
Very fair	3 (1.3)	5 (1.5)		13 (6.7)	16 (6.0)	
Fair	94 (41.6)	169 (49.4)		76 (39.0)	96 (35.8)	
Olive	87 (38.5)	116 (33.9)		28 (14.4)	42 (15.7)	
Light brown	36 (15.9)	47 (13.7)		64 (32.8)	102 (38.1)	
Dark brown or very dark	3 (1.3)	4 (1.2)		13 (6.7)	11 (4.1)	
Geographic location (*n*, %)			n/a			0.83
Tampa	226 (100)	342 (100)		93 (47.7)	124 (46.3)	
Ponce	0	0		102 (52.3)	144 (53.7)	
Language preference (*n*, %)			n/a			1.00
English	226 (100)	342 (100)		85 (43.6)	116 (43.3)	
Spanish	0	0		110 (56.4)	152 (56.7)	
Season baseline survey completed (*n*, %)						
Spring			0.90			0.36
Summer	66 (29.2)	93 (27.2)		61 (31.3)	93 (34.7)	
Fall	38 (16.8)	63 (18.4)		32 (16.4)	32 (11.9)	
Winter	60 (26.5)	96 (28.1)		62 (31.8)	77 (28.7)	
Competed post-intervention telephone call (*n*, %)	148 (75.9)	199 (74.5)	0.82	184 (81.4)	254 (74.3)	**0.06**
**Skin cancer-related behaviors over the past year**						
Sun exposure						
Weekday, range 0–6 h (mean, SD)	1.4 (1.0)	1.4 (1.3)	0.19	1.58 (1.52)	1.63 (1.57)	0.63
Weekend, range 0–6 h (mean, SD)	2.4 (1.4)	2.4 (1.6)	0.61	1.86 (1.53)	2.03 (1.63)	0.32
Sunburns, range 0–5 or more (mean, SD)	0.7 (1.0)	0.8 (1.1)	0.28	0.57 (0.99)	0.56 (1.0)	0.71
Outdoor intentional tanning (mean, SD)	2.0 (0.9)	2.0 (1.0)	0.63	1.71 (0.92)	1.60 (0.79)	0.34
Indoor tanning (*n*, %)	12 (5.3)	7 (2.0)	**0.05**	6 (3.1)	7 (2.7)	0.78
Wearing a hat often or always (*n*, %)	56 (25.2)	95 (28.0)	0.53	32 (16.4)	39 (14.6)	0.70
Wearing a shirt with sleeves often or always (*n*, %)	154 (69.1)	228 (66.9)	0.65	123 (63.1)	158 (59.0)	0.47
Wearing sunglasses often or always (*n*, %)	153 (68.6)	233 (68.5)	1.00	89 (45.6)	116 (43.3)	0.70
Using sunscreen often or always (*n*, %)	66 (29.6)	127 (37.2)	**0.07**	39 (20.0)	62 (23.1)	0.47
Seeking shade or using an umbrella often or always (*n*, %)	71 (31.8)	106 (31.2)	0.94	87 (44.6)	108 (40.3)	0.42
Total number of sun protection behaviors practiced often or always, range 1–5 (median, IQR)	2 (2)	2 (2)	0.51	2 (2)	2 (2)	0.34
**Skin cancer-related distress**						
Recent worry over the past two weeks (*n*, %)			1.00			0.25
Rarely or never	211 (93.4)	321 (93.9)		160 (82.1)	205 (76.5)	
Sometimes, often, or all the time	14 (6.2)	20 (5.8)		34 (17.4)	59 (22.0)	
Recent concern over the past two weeks (*n*, %)			0.50			0.85
Not at all concerned	192 (85.0)	299 (87.4)		140 (71.8)	193 (72.0)	
A bit concerned, concerned, or very concerned	33 (14.6)	42 (12.3)		54 (27.7)	70 (26.1)	
Worry scale, range 1–5 (median, IQR)	1.67 (0.67)	1.67 (0.67)	0.74	1.67 (1)	1.67 (1)	0.87
Recent distress over the past week, range 0–75 (median, IQR)	0 (5)	0 (5)	0.95	1 (10)	1 (10)	0.86
Avoidance subscale, range 0–40 (median, IQR)	0 (5)	0 (4)	0.69	0 (6)	0 (7)	0.79
Intrusion subscale, range 0–35 (median, IQR)	0 (1)	0 (1)	0.85	0 (3)	0 (3)	0.88
**Skin cancer-related beliefs**						
Absolute risk of developing skin cancer (*n*, %)			0.87			0.46
Unlikely	86 (38.1)	131 (38.3)		79 (40.5)	123 (45.9)	
Likely	31 (13.7)	52 (15.2)		63 (32.3)	78 (29.1)	
No idea	108 (47.8)	158 (46.2)		52 (26.7)	63 (23.5)	
Comparative risk of developing skin cancer (*n*, %)			0.48			0.68
Well below average	23 (10.2)	30 (8.8)		41 (21.0)	44 (16.4)	
Below average	59 (26.1)	93 (27.2)		50 (25.6)	67 (25.0)	
Average	118 (52.2)	167 (48.8)		72 (36.9)	121 (45.1)	
Above or well above average	25 (11.1)	51 (14.9)		31 (15.9)	29 (10.8)	
Perceived severity, range 1–4 (median, IQR)	2.6 (0.4)	2.6 (0.4)	0.93	2.43 (0.43)	2.43 (0.43)	0.29
Response efficacy, range 1–4 (median, IQR)	3.3 (0.9)	3.3 (0.7)	0.57	3.57 (1.0)	3.43 (0.86)	**0.08**
Self-efficacy, range 1–4 (median, IQR)	3.6 (0.7)	3.4 (0.9)	**0.02**	3.57 (0.86)	3.43 (0.71)	**0.01**

**Bold values** indicate variables included in multivariable modeling of intervention effects at the first follow-up. ^a^ Participants who responded “educated outside the USA” were coded to the median education level (some college). ^b^ not applicable. ^c^ *p*-value reflects comparison of Puerto Rican versus all other Hispanic identities.

**Table 2 cancers-16-04027-t002:** Multivariate-adjusted ^a^ differences in intervention effects at the first follow-up on skin cancer-related behaviors, distress, and beliefs between participants at higher risk and those at average risk.

	Non-Hispanic White	Hispanic
Outcome	β (95% CI) or OR (95% CI)	β (95% CI) or OR (95% CI)
**Skin cancer-related behaviors**		
Sun exposure		
Weekday hours	−0.13 (−0.30, 0.04)	0.19 (−0.10, 0.49)
Weekend hours	**−0.25 (−0.46, −0.04)**	−0.16 (−0.46, 0.14)
Sunburns	0.09 (−0.02, 0.21)	0.0 (−0.16, 0.15)
Outdoor intentional tanning	−0.03 (−0.15, 0.10)	0.02 (−0.14, 0.18)
Indoor tanning ^b^	0.47 (0.09, 2.03)	2.54 (0.24, 126.8)
Wearing a hat	0.99 (0.52, 1.86)	0.97 (0.35, 2.68)
Wearing a sleeved shirt	1.05 (0.63, 1.75)	0.94 (0.50, 1.79)
Wearing sunglasses	1.15 (0.62, 2.13)	1.02 (0.51, 2.05)
Using sunscreen	1.24 (0.72, 2.13)	1.42 (0.63, 3.21)
Seeking shade or using an umbrella	1.31 (0.79, 2.17)	1.39 (0.73, 2.64)
Total number of sun protection behaviors	0.09 (−0.11, 0.29)	0.12 (−0.16, 0.39)
**Skin cancer-related distress**		
Recent worry over the past two weeks	1.99 (0.78, 5.09)	1.91 (0.87, 4.22)
Recent concern over the past two weeks	1.61 (0.80, 3.26)	2.01 (0.84, 4.8)
Worry scale	**0.09 (0.01, 0.18)**	−0.04 (−0.22, 0.13)
Recent distress over the past week	1.21 (−0.23, 2.64)	−0.45 (−3.20, 2.30)
Avoidance subscale	0.67 (−0.37, 1.70)	0.35 (−1.49, 2.19)
Intrusion subscale	0.52 (−0.01, 1.05)	−0.20 (−1.47, 1.08)
**Skin cancer-related beliefs**		
Absolute risk of developing skin cancer		
Unlikely	Ref	Ref
Likely	**2.16 (1.10, 4.22)**	**3.40 (1.40, 8.27)**
No idea	**1.73 (1.04, 2.87)**	1.86 (0.93, 3.74)
Comparative risk of developing skin cancer	**0.40 (0.24, 0.57)**	**0.50 (0.28, 0.73)**
Perceived severity	0.05 (−0.02, 0.12)	−0.07 (−0.17, 0.03)
Response efficacy	**0.10 (0.01, 0.19)**	0.01 (−0.12, 0.15)
Self-efficacy	0.03 (−0.05, 0.12)	−0.01 (−0.13, 0.11)

**Bold values** indicate a statistically significant measure of effect. ^a^ Effects were adjusted for variables measured at the baseline with differences (*p* ≤ 0.10) between risk groups, the outcome measured at baseline, predictors of missingness at the first follow-up, and predictors of outcome at the first follow-up. ^b^ Due to small numbers, reported here are conditional maximum likelihood estimates of the odds ratio with Fisher’s exact confidence intervals and *p*-values comparing ever having indoor tanning reported at the first follow-up between participants at higher risk and those at average risk.

**Table 3 cancers-16-04027-t003:** Multivariate-adjusted ^a^ differences in intervention effects at the second follow-up on skin cancer-related behaviors, distress, and beliefs between participants at higher risk and those at average risk.

	Non-Hispanic White	Hispanic
Outcome	β (95% CI) or OR (95% CI)	β (95% CI) or OR (95% CI)
**Skin cancer-related behaviors**		
Sun exposure		
Weekend hours	−0.03 (−0.26, 0.20)	
**Skin cancer-related distress**		
Worry scale	0.06 (−0.02, 0.15)	
**Skin cancer-related beliefs**		
Absolute risk of developing skin cancer		
Unlikely	Ref	Ref
Likely	**2.25 (1.35, 3.75)**	1.87 (0.85, 4.10)
No idea	**3.35 (1.70, 6.62)**	**2.22 (1.07, 4.60)**
Comparative risk of developing skin cancer	**0.49 (0.33, 0.65)**	**0.42 (0.17, 0.67)**
Response efficacy	0.05 (−0.03, 0.13)	

**Bold values** indicate a statistically significant measure of effect. ^a^ Only variables with statistically significant effects at the first follow-up were analyzed for effects at the second follow-up. Effects were adjusted for variables measured at the baseline with differences (*p* ≤ 0.10) between risk groups, the outcome measured at the baseline, predictors of missingness at the second follow-up, and predictors of outcome at the second follow-up.

## Data Availability

The data presented in this study are available on request from the corresponding author.

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
