# Peer review of "Changes in Skin Cancer-Related Behaviors, Distress, and Beliefs in Response to Receipt of Low- to Moderate-Penetrance Genetic Test Results for Skin Cancer Risk"

_cancers, 2024, doi:10.3390/cancers16234027_

Round 1
Reviewer 1 Report
Comments and Suggestions for Authors
Major issues
1. It looks like people whose genetic risk score was high had better behaviors at baseline in NHW group than average risk. P. 5 lines 227-230. You state that those with average risk have less healthy behaviors. However, you can also word this as those with high risk had more healthy behaviors at baseline. Thus, any new reports of the genetic risk may have less of an effect. In the discussion, you state that these differences were controlled for in the regression analysis (Page 10, lines 345-347). Please discuss this more in the discussion.
Minor issues
1. Page 1, line 38. What is “average comparative chance in getting skin cancer”? This term is confusing and really is just their perception of risk. Please clarify or define early.
2. Page 2, line 84. “Carcinoma, and many” comma should be semicolon.
3. Page 9, line 285. “cancer-related beliefs, although …” comma should be semicolon.
Author Response
Comment 1. It looks like people whose genetic risk score was high had better behaviors at baseline in NHW group than average risk. You state that those with average risk have less healthy behaviors. However, you can also word this as those with high risk had more healthy behaviors at baseline. Thus, any new reports of the genetic risk may have less of an effect. In the discussion, you state that these differences were controlled for in the regression analysis. Please discuss this more in the discussion.
Response 1. We thank the reviewer for the opportunity to expand our discussion about intervention effects on cancer-related behaviors among participants at MC1R higher risk. We have now added the following text to the discussion section (lines 338-341):
“Interestingly, although NHW participants at MC1R higher risk already reported statisti-cally less indoor tanning and more frequent use of sunscreen at baseline than did NHW participants at MC1R average risk, we still observed a positive intervention effect on these two skin cancer prevention behaviors comparing these two risk groups.”
Comment 2. What is “average comparative chance in getting skin cancer”? This term is confusing and really is just their perception of risk. Please clarify or define early.
Response 2. We have rewritten this sentence (lines 37-38) in the abstract for clarity, and it now reads: “On average, higher risk NHW and Hispanic participants reported persistent increased risk of getting skin cancer compared to similar others.” We also have taken this opportunity to edit other descriptions of comparative risk throughout the manuscript to enhance clarity, including replacing the word ‘chance’ with ‘risk’ (line 213, line 220, Table 1, Table 2, lines 264-267, Table 3, lines 278-281, lines 283-285, line 287).
Comment 3. “Carcinoma, and many” comma should be semicolon.
Response 3. We have made this correction in the text (line 84).
Comment 4. “cancer-related beliefs, although …” comma should be semicolon.
Response. 4. We have made this correction in the text (line 297).
Reviewer 2 Report
Comments and Suggestions for Authors
The abstract is prepared accordingly.
The introduction is sufficient, refers to relevant literature. On the other hand, the explicitly described purpose of the study is missing - citation correct should be completed.
Material and methods - not fully understood that:
Information on the setting, participants, research surveys and study measures, genetic testing, and precision prevention materials of the two parent intervention studies, one among non-Hispanic White (NHW) participants and one among Hispanic participants, have been described in detail elsewhere [8,9]. - While I understand that this is a reference to other publications, information about participants, inclusion and exclusion criteria, and other important methodological data should also be included in this manuscript. Please explain in more detail if there were differences between the studies on each of the 2 populations (whites versus Hispanics) and why these studies differed, since the assumption was that the populations would be comparable.
The results are presented in a clear and transparent manner. I would suggest a possible modification of Table 1, which is quite elaborate, perhaps some of the results can be described in the text.
The discussion includes references to the topic of the study. I suggest expanding it to include a comparison of studies on skin cancer prevention in other populations, such as European ones.
The conclusions are formulated correctly. The conclusion about providing important information for public health genomics is questionable. By definition, genomics is a scientific field that deals with the analysis of the genome of organisms. This study deals with skin cancer prevention. Perhaps this is a linguistic error.
Author Response
Comment 1. The introduction is sufficient, refers to relevant literature. On the other hand, the explicitly described purpose of the study is missing - citation correct should be completed.
Response 1. We believe the final paragraph of the Introduction (lines 95-99) clearly conveys the purpose of our current study. This paragraph reads:
"In the secondary data analyses reported here, we compared the effect of the precision prevention on skin cancer-related behaviors, distress, and beliefs, among participants at MC1R higher risk and MC1R average risk at the first of two follow-up surveys. We then evaluated the durability of any statistically significant difference by comparing the effect at the second follow-up."
Comment 2. While I understand that this is a reference to other publications, information about participants, inclusion and exclusion criteria, and other important methodological data should also be included in this manuscript. Please explain in more detail if there were differences between the studies on each of the 2 populations (whites versus Hispanics) and why these studies differed, since the assumption was that the populations would be comparable.
Response 2. We have added the following additional information about recruitment, inclusion, and exclusion criteria on the two parent studies in section 2.1 Setting and Participants (lines 105-117).
Participants who self-reported being non-Hispanic, White, fluent in English, and 18 years of age or older, and who reported having few skin cancer risk phenotypes based on ability to tan, tendency to burn, hair color, and freckling were recruited from primary care clinics in Tampa, Florida between September 2015 and September 2018. NHW individuals who had a full-body skin examination within the past year, or a personal history of melanoma were ineligible for study participation. Participants who self-reported being Hispanic, fluent in either English or Spanish, and 18 (Tampa) or 21 (Ponce) years of age or older were recruited from primary care settings in Tampa, Florida and Ponce, Juana Díaz, and Salinas, Puerto Rico between September 2018 and January 2020. Hispanic individuals who had a full-body skin examination within the past year, a personal history of melanoma, or a personal history of more than one basal cell carcinoma and/or squamous cell carcinoma were ineligible for study participation.
Comment 3. I would suggest a possible modification of Table 1, which is quite elaborate, perhaps some of the results can be described in the text.
Response 3. We agree that Table 1 is long, but we feel this format is the most efficient and clear way to present the wealth of baseline data describing our two study populations. Moving results from the Table to text would result in unnecessary lengthening of the manuscript body.
Comment 4. The discussion includes references to the topic of the study. I suggest expanding it to include a comparison of studies on skin cancer prevention in other populations, such as European ones.
Response 4. In our Discussion, we purposely limited comparison of our results to other studies also using a low-moderate genetics precision prevention approach to affect change in skin cancer-related behaviors, skin cancer-related distress, and skin cancer-related beliefs. We are unaware of published findings from such precision prevention studies conducted in Europe.
Because the available literature on skin cancer prevention interventions is expansive, dates back several decades, and uses diverse approaches, we purposely limited comparison of our results to other studies that have a common goal to our research, which is incorporating a genetic-based precision prevention approach to study changes in skin cancer-related behaviors, skin cancer-related distress, and skin cancer-related beliefs. We do compare our results to an available and comparable precision prevention intervention study set in Australia that provided participants with risk estimates derived from a polygenic risk score, and we clarified this genetic feedback in our revised text (line 343). We are unaware, however, of comparable published findings from skin cancer genetic precision prevention studies conducted in Europe.
Comment 5. The conclusion about providing important information for public health genomics is questionable. By definition, genomics is a scientific field that deals with the analysis of the genome of organisms. This study deals with skin cancer prevention. Perhaps this is a linguistic error.
Response 5. The reviewer is correct that genomics is a specific scientific field. However, there also is scientific field called public health genomics that seeks to translate scientific findings from the genomic sciences to inform population health in an effective and responsible fashion. This framework is conveyed in the opening sentence of our Introduction. There we provide a full working definition of public health genomics, referencing the publication by Khoury et al. We believe our study and its findings clearly fall within the realm of public health genomics.